# An RBD-Based Diagnostic Method Useful for the Surveillance of Protective Immunity against SARS-CoV-2 in the Population

**DOI:** 10.3390/diagnostics12071629

**Published:** 2022-07-05

**Authors:** Dolores Adriana Ayón-Núñez, Jacquelynne Cervantes-Torres, Carlos Cabello-Gutiérrez, Sergio Rosales-Mendoza, Diana Rios-Valencia, Leonor Huerta, Raúl J. Bobes, Julio César Carrero, René Segura-Velázquez, Nora Alma Fierro, Marisela Hernández, Joaquín Zúñiga-Ramos, Gerardo Gamba, Graciela Cárdenas, Emmanuel Frías-Jiménez, Luis Alonso Herrera, Gladis Fragoso, Edda Sciutto, Francisco Suárez-Güemes, Juan Pedro Laclette

**Affiliations:** 1School of Veterinary Medicine and Zootechnics, Universidad Nacional Autónoma de México, Coyoacán, Ciudad de México 04510, Mexico; d.adrianaayon@gmail.com (D.A.A.-N.); rsegura@fmvz.unam.mx (R.S.-V.); 2Biomedical Research Institute, Universidad Nacional Autónoma de México, Coyoacán, Ciudad de México 04510, Mexico; jcervantes@iibiomedicas.unam.mx (J.C.-T.); dagariva10@gmail.com (D.R.-V.); leonorhh@iibiomedicas.unam.mx (L.H.); rbobes@iibiomedicas.unam.mx (R.J.B.); carrero@iibiomedicas.unam.mx (J.C.C.); noraalma@iibiomedicas.unam.mx (N.A.F.); marysel_01@yahoo.com.mx (M.H.); gamba@iibiomedicas.unam.mx (G.G.); gladis@unam.mx (G.F.); edda@unam.mx (E.S.); 3Instituto Nacional de Enfermedades Respiratorias Ismael Cosío Villegas, Calzada de Tlalpan 4502, Belisario Domínguez Secc. 16, Tlalpan, Ciudad de México 14080, Mexico; carloscginer@gmail.com (C.C.-G.); joaquin.zuniga@iner.gob.mx (J.Z.-R.); 4Laboratorio de Biofarmacéuticos Recombinantes, Facultad de Ciencias Químicas, Universidad Autónoma de San Luis Potosí, Av. Dr. Manuel Nava 6, San Luis Potosí 78210, Mexico; rosales.s@uaslp.mx; 5Instituto Nacional de Ciencias Médicas y Nutrición Salvador Zubirán, Vasco de Quiroga 15, Belisario Domínguez Secc. 16, Tlalpan, Ciudad de México 14080, Mexico; 6Instituto Nacional de Neurología y Neurocirugía, Av. Insurgentes Sur 3877, Tlalpan, Ciudad de México 14269, Mexico; gracielacardenas@yahoo.com.mx; 7Instituto Nacional de Medicina Genómica, Periférico Sur 4809, Ciudad de México 14610, Mexico; jfrias@inmegen.gob.mx (E.F.-J.); herreram@iibiomedicas.unam.mx (L.A.H.)

**Keywords:** COVID-19, receptor-binding domain (RBD), serologic diagnosis, ELISA, neutralizing antibodies

## Abstract

After more than two years, the COVID-19 pandemic is still ongoing and evolving all over the world; human herd immunity against SARS-CoV-2 increases either by infection or by unprecedented mass vaccination. A substantial change in population immunity is expected to contribute to the control of transmission. It is essential to monitor the extension and duration of the population’s immunity to support the decisions of health authorities in each region and country, directed to chart the progressive return to normality. For this purpose, the availability of simple and cheap methods to monitor the levels of relevant antibodies in the population is a widespread necessity. Here, we describe the development of an RBD-based ELISA for the detection of specific antibodies in large numbers of samples. The recombinant expression of an RBD-poly-His fragment was carried out using either bacterial or eukaryotic cells in in vitro culture. After affinity chromatography purification, the performance of both recombinant products was compared by ELISA in similar trials. Our results showed that eukaryotic RBD increased the sensitivity of the assay. Interestingly, our results also support a correlation of the eukaryotic RBD-based ELISA with other assays aimed to test for neutralizing antibodies, which suggests that it provides an indication of protective immunity against SARS-CoV-2.

## 1. Introduction

The ongoing pandemic of COVID-19 [1] has not only provoked a global health emergency but also modified numerous social behaviors and impacted the economy of the whole world [2]. Since the beginning of the COVID-19 pandemic, researchers around the world began to explore alternatives for diagnosis, vaccination, and treatment. Until now, the total number of infections in the world has reached 514 million people, resulting in over 6.1 million deaths (http://systems.jhu.edu/research/public-health/ncov/jhu.edu accessed on 2 May 2022). Over eleven billion vaccine doses have been administered all over the world, but the pandemic appears far from ending, currently being driven by the Omicron variant [3,4]. Although vaccination does not act as full protection against infection, it does reduce the transmission of the virus by vaccinated people. Of utmost importance, it also reduces the severity of symptoms and the possibility of developing a productive infection in the vaccinated–infected people [5,6,7].

Despite impressive progress made in record time, there are still many aspects to be learned. Of special importance is the knowledge of how long the immunity induced by the different vaccines currently in use will last, including for COVID-19-recovered individuals. The drop in serum antibodies after vaccination, together with the increase in severe cases and deaths in vaccinated people will be indicators of the need to improve immunity against the disease. This will be especially important for individuals with comorbidities who are more likely to develop fatal cases [8]. Therefore, the continuous epidemiological surveillance of vaccinated people will be required to aid vaccination programs in the near future. For this purpose, simple methods are necessary to quantify the levels of antibodies induced against the virus. Among induced antibodies, it is critically relevant to determine those with neutralizing activity capable of blocking the entry of the virus into the cell [9].

The main target of neutralizing antibodies induced by vaccination is the SARS-CoV-2 spike protein (S). This protein has a cell receptor-binding domain (RBD), which is the main determinant of virus entrance into host cells, as well as the critical target of neutralizing antibodies [10]. The receptor in the host cell is the angiotensin-converting enzyme 2 (ACE2) located on the cell membrane [11,12].

A number of methods directed to detect different viral proteins are available for the evaluation of the humoral immune response of individuals against SARS-CoV-2, including the S and nucleocapsid (N) proteins [13,14]. Some of them achieve high sensitivity and specificity and have been widely used for the identification of COVID-19 patients. The detection of antibodies directed to the S protein and particularly to the RBD fragment provides the most pertinent information about the immune protective status of individuals, as this moiety is involved in the penetration of viral particles into the target cells [15,16]. Here, we report a simple and quick method for the detection of SARS-CoV-2 antibodies using purified recombinant eukaryotic RBD fragments, which correlate with the presence of neutralizing, protective immunity in human individuals. To develop this tool, we tested two recombinant RBD products, a bacterial RBD fragment of 194 residues, and the eukaryotic complete 223 amino acids RBD. The ability of antibodies to recognize the RBD by ELISA has been shown to correlate with their ability to neutralize the virus in vitro [17,18]. This type of assay does not require the use of the complete virus, can be carried out in laboratories with biosafety level 2 (BSL-2), and has been extensively evaluated in pre-pandemic and pandemic sera from different hospitals in Mexico City.

## 2. Materials and Methods

The recombinant expression and purification of SARS-CoV-2 RBD in *Escherichia coli*.

The plasmid pET-22b(+), containing the SARS-CoV-2 S protein RBD sequence (YP_009724390.1; Asn331-Val524 residues), coding for a fragment of 194 residues, was introduced into competent *Escherichia coli* BL21(DE3) cells (Stratagene, San Diego, CA, USA) by electroporation. Transformed clones were induced to expression for 2, 4, and 6 h at 30 °C by incubation with 1 mM IPTG. Cells were collected by centrifugation and dissolved in lysis buffer (50 mM Tris-HCl pH 8.0, 100 mM NaCl, and 1 mM EDTA) and the insoluble fraction was solubilized with a buffer of 20 mM Tris-HCl pH 8.0, 6 M Urea, 2 mM EDTA, and 50 mM DTT. The RBD-His tag fragment was purified from this fraction using affinity chromatography in an ÄKTA prime plus system (GE Healthcare Life Sciences, Piscataway, NJ, USA) in elution buffer (Na_2_HPO_4_ 50 mM, NaCl 0.3 M, Imidazole 250 mM, and 6 M urea). Afterward, it was dialyzed in a buffer containing Na_2_HPO_4_ 50 mM, NaCl_2_ 0.3 M, and 3 M urea. Protein concentrations were determined by Lowry’s method [19]. The total bacterial extracts and the purified fractions were run on 12% SDS-PAGE. Finally, the purified fraction was transferred to a PVDF membrane to determine immune recognition by Western blot. The membranes were blocked with 3% BSA in PBS-Tween 0.05% (PBS-T) for 2 h and incubated with a primary anti-poly-His antibody for 2 h (Roche Diagnostics, Basel, Switzerland), diluted 1:4000, then washed with PBS-T and incubated for 1 h with secondary HRP-conjugated anti-mouse IgG polyclonal antibody (Sigma, St. Louis, MI, USA) diluted 1:2500. The reaction was revealed using 3 mg/mL of 3,3-diaminobenzidine (Sigma) in PBS and 30% hydrogen peroxide.

The recombinant expression and purification of SARS-CoV-2 RBD in HEK293T mammal cells.

The plasmid pCAGGS containing the SARS-CoV-2 (Wuhan-Hu-1) spike glycoprotein receptor-binding domain (RBD_319–541_) (Cat. NR-52309 BEI Resource), coding for the full 223 amino acid residues RBD fragment, was expanded in the *E.coli* HST08 strain (Takara Bio, Kusatsu, Japan) and transfected into human embryonic kidney (HEK) 293 cells (Thermo Fisher Scientific, Waltham, MA, USA). Approximately 24 h before carrying out the transfection, 3 × 10^6^ cells per mL were cultured in DMEM high glucose (GIBCO, Waltham, MA, USA) with 10% FBS (ByProducts), 2 mM L-Glutamine, and 1% Penicillin-Streptomycin (GIBCO) at 37 °C and 5% CO_2_ atmosphere until reaching 80% confluence. Before transfection, the culture medium was replaced with fresh DMEM supplemented with 2% FBS. HEK293 cells were transfected using the PolyJet™ reagent (SignaGen, Frederick, MD, USA) in a DNA:Polyjet ratio of 3:1, following the manufacturer’s instructions. After 12 h, the medium was replaced with 12 mL of DMEM medium with 10% FBS. The culture supernatants were harvested 5, 7, and 9 days after incubation and then clarified by centrifugation at 2000× *g* for 20 min at 4 °C. The supernatant was filtered through a 0.45 μm membrane and the recombinant RBD fragment (rRBD) was purified by gravity-flow affinity chromatography using Ni-NTA resin (Qiagen, Hilden, Germany). The rRBD-containing fractions were pooled and dialyzed to remove imidazole using an Na_2_HPO_4_ 50 mM–NaCl 0.3 M buffer and concentrated in a 50K Amicon Ultra Centrifugal Filter Unit (Merck, Kenilworth, NJ, USA). The protein content was quantified by UV/VIS spectrophotometry on a Nanodrop 2000 (Thermo Scientific) and by Lowry´s method. The characterization of the rRBD included SDS-PAGE analysis and Western blot testing, as described above.

### 2.1. Sera Samples

A total of 165 sera from patients with COVID-19 (collected between March and June 2020) were obtained from two hospitals in Mexico City (the National Institute of Respiratory Diseases (INER) and the National Institute of Medical Sciences and Nutrition (INCMN)). All enrolled patients were confirmed to be infected with SARS-CoV-2 by real-time RT-PCR. For the optimization of the RBD assay, a total of 16 sera from COVID-19 convalescent individuals were collected at least 25 days and one month after the onset of symptoms at the National Institute of Neurology and Neurosurgery (INNN). Forty negative sera samples were obtained from the INCMN from patients admitted in 2014, before the COVID-19 pandemic. All the sera were heat-inactivated at 56 °C for 1 h and stored at −20 °C before use [16]. All individuals that provided blood samples signed an informed consent letter. The anonymity of all blood sample donors was assured. Protocols were authorized by Bioethical Committees at INNN (permission No. 125/20) and INER (permission No. B0920).

### 2.2. ELISA Testing of Sera

Different conditions were assayed with the purified bacterial and eukaryotic rRBD. After initial ELISA testing, our results showed that the amount of bacterial rRBD to produce clear recognition differences between the sera from healthy and infected individuals was far higher than for eukaryotic rRBD. Therefore, the ELISA testing of serum samples was as follows: flat-bottom 96-well plates (Nunc MaxiSorp™ ThermoFisher Scientific) were coated overnight at 4 °C with 50 µL per well containing 5 µg of bacterial rRBD or 50 ng of eukaryotic rRBD, both in carbonate buffer 0.05 M, pH 9.6. The next day, the plates were washed three times with PBS-T and blocked with 150 µL per well of 3% skim milk prepared in PBS for 2 h at room temperature (RT = 25 °C). Afterward, the plates were washed, and 50 µL of the sera samples diluted at 1:200 in PBS-T containing 1% skim milk were added. Duplicates were incubated for 2 h at RT. Next, the plates were washed four times with PBS-T and a goat anti-human IgG-horseradish peroxidase (Abcam, Cambridge, UK), diluted 1:20,000, was added to each well, and incubation proceeded for 45 min. The plates were washed and the well´s content was completely removed to add the TMB Substrate Reagent (ThermoFisher Scientific) for 20 (bacterial rRBD) or 12 (eukaryotic rRBD) minutes, before the reaction was stopped by the addition of 0.2 M sulphuric acid. The optical density at 450 nm (OD450 nm) was measured using a Sinergy (BioTek, Winooski, VT, USA) plate reader. As a reference, our ELISA using eukaryotic RBD fragments was compared with one commercially available kit for the diagnosis of SARS-CoV-2, which also uses the RBD fragment as an antigen (Kit UDITEST-V2G^®^).

### 2.3. Microneutralization Assays

Microneutralization (MN) assays were carried out in two-fold serial dilutions in EMEM, of 40 sera samples from vaccinated individuals (pre-selected after antibody testing using our eukaryotic rRBD ELISA), added to an equal volume (150 µL) of SARS-CoV-2 virus at an MOI of 0.1. The virus, isolated from the nasopharyngeal swab of a COVID-19 patient, was sequenced and the sequence was compared with those reported in the GenBank, showing 100% identity with the USA/CO-CDPHE-2100177494/2020 (GenBank: ON228044.1). The antibody/virus mixture was incubated at 37 °C for 1 h and then added to VERO E6 cell monolayers for 3 days at 37 °C under an atmosphere of 5% CO_2_. The resulting cytopathic effect (CPE) was visualized every day under the optical microscope. The cells were washed with PBS and fixed with ethanol:acetone (1:1) for 15 min and stained with violet crystal for 20 min. Positive (with a pre-evaluated lytic capacity) and negative controls (no virus added) were included in the assay. Antibody titers were expressed as the maximal dilution at which the serum inhibited CPE. Three ranges of antibody response were defined: the low range corresponds to the sera under the cut off (media of negative sera plus three standard deviations), medium reaches up to 50% of the maximal OD obtained in all sera samples, and high corresponds to the titers over the later. This procedure was carried out in a BSL3 laboratory.

### 2.4. Statistical Analysis

We defined a sample as positive if the OD450 value was three standard deviations (SD) above the mean of a panel of at least five negative controls. The correlation between the sera results of ELISA and MN assays after the logarithmic transformation of MN titers was assessed using Pearson´s correlation analysis.

### 2.5. Results

#### 2.5.1. Expression and Purification of Bacterial rRBD

The BL21(DE3) cells transformed with the pET-22b(+) plasmid, inserted with an RBD coding sequence and induced with IPTG, resulted in high expression levels of a 25 kDa recombinant product, as expected for the RBD (194 amino acid residues) fragment, reaching a maximal level of expression at 4 h post-induction (Figure 1A). After the purification of the recombinant product by FPLC in a nickel column, at least four protein fractions were obtained, of which the second peak showed more than 90% purity as judged by SDS-PAGE analysis using coomassie blue staining (Figure 1B). The recombinant fragment was clearly recognized by a commercial anti-poly-His antibody by Western blot (Figure 1C).

The antigenicity evaluation of bacterial rRBD using the human sera of healthy and infected individuals.

A total of 62 sera obtained from COVID-19 patients from different hospitals were evaluated by ELISA against the bacterial rRBD. From these, 49 sera were positive (79%) according to the cut-off previously determined (Figure 2). All negative serum samples (100%) did not react against this RBD protein.

#### 2.5.2. Expression and Purification of Eukaryotic rRBD

The transfection of HEK293T mammalian cells with the Polyjet reagent was successful and resulted in manageable levels of expression (Figure 3A). Five days after transfection, the cell supernatants were collected and pooled for the purification of the recombinant product through a nickel column and exhaustively dialyzed to remove the imidazole (Figure 3B). Affinity-purified eukaryotic rRBD was recognized by an anti-poly-His antibody by Western blot, as a protein band at ~30 kDa, indicating that monomeric eukaryotic rRBD was the most common material (Figure 3C).

The antigenicity evaluation of eukaryotic rRBD using human sera of healthy and infected individuals.

For practical reasons, a larger number of COVID-19 patients and pre-pandemic sera were used to determine the antibody response against SARS-CoV-2 infection using the eukaryotic rRBD. ELISA testing results are shown in Figure 4. The reactivity of the positive sera reached OD450 values ranging from 0.4 to 2.0, above the negative threshold, in 153 out of the 165 samples, thus increasing the sensitivity of the assay to 93%. All negative serum samples (100%) did not react against this RBD protein.

#### 2.5.3. Comparison between the Bacterial and the Eukaryotic rRBD-Based ELISA

To further compare the sensitivity of both bacterial and eukaryotic rRBD assays, a total of 30 positive and 14 negative serum samples previously tested by RT-PCR were assessed in parallel. Figure 5 shows that the bacterial rRBD detected a total of 24 positive IgG sera while the eukaryotic rRBD detected a total of 27 positive, reaching sensitivities of 80 and 90%, respectively.

#### 2.5.4. Correlation of SARS-CoV-2 Neutralization Activity and Anti-RBD IgG Antibody Levels

We carried out a comparison between the anti-RBD IgG antibody levels and the neutralizing response evaluated in an MN assay (Figure 6). Samples from low, medium, and high antibody level sera from vaccinated individuals (pre-selected after antibody testing using our eukaryotic rRBD ELISA) were evaluated for their neutralization titers. The sera were tested using 2-fold serial dilutions. Five pre-pandemic control sera, used as negative controls, were also tested for neutralization activity. Results showed that control sera were neutralized only at a 1:2 dilution, whereas sera from the vaccinated group efficiently neutralized the virus from a 1:4 up to a 1:1024 dilution. Therefore, the anti-rRBD IgG levels detected in our ELISA closely are correlated with the SARS-CoV-2 neutralization ability according to our in vitro assay (Pearson’s *r* = 0.87, *p* < 0.0001).

#### 2.5.5. Comparison of the Eukaryotic rRBD vs. Commercial SARS-CoV-2 ELISA

Our method was compared with a commercially available kit for the diagnosis of SARS-CoV-2 which also uses the RBD moiety as an antigen for detection (Kit UDITEST-V2G^®^). This ELISA kit exhibited 99% sensitivity with sera collected 15 days after the onset of COVID-19 symptoms. A panel of 75 human sera from infected and uninfected individuals was used for comparison with the kit. The diagnostic sensitivity was calculated according to the formula SN = TP/(TP + FN) ∗ 100, where TP and FN indicate true-positive and false-negative, respectively. Eleven FN sera were found when our ELISA was compared against the UDITEST kit, resulting in a decrease in sensitivity to 82.3%.

## 3. Discussion

The spike (S) glycoprotein comprises the subunits S1 and S2, which are responsible for binding to host cell receptors and the fusion of the virus and host-cell membranes to mediate the penetration of coronavirus particles. The S1 subunit contains the receptor-binding domain (RBD), which interacts directly with the angiotensin-converting enzyme 2 (ACE2) in the target cell [20,21]. As RBD is the moiety of the S protein involved in the penetration of SARS-CoV-2 into the target cells, using this region as an antigen favored the recognition of antibodies with neutralizing activity [22]. In fact, several SARS-CoV-2 serology ELISA protocols are now based on the RBD [14,16]. However, although these kits are commercially available, they are expensive when used in large numbers of samples to monitor the antibody response in the vaccinated population.

In this study, an ELISA protocol based on the use of SARS-CoV-2 rRBD is reported. Two RBD recombinant products were produced using two different platforms. A 194 amino acid fraction of RBD was expressed in *E. coli* prokaryote cells, whereas a complete 223 amino acid RBD was produced in a eukaryotic immortalized cell line derived from human embryonic kidney cells (HEK293T). Both rRBDs were used to develop ELISA tests for the detection of anti-RBD IgG antibodies.

These assays were carried out using a set of 165 sera of patients that required hospitalization and were RT-PCR confirmed for COVID-19 and 40 sera of negative individuals collected previous to the COVID-19 pandemic. All sera were obtained from two national institutes of health in Mexico City. The assay demonstrated higher sensitivity for the eukaryotic rRBD when compared to the prokaryotic counterpart, even using a 100-fold lower amount of protein for sensibilization (90% vs. 80%, respectively). Differences might result from the change in the length of the sequence (Appendix A) and from expected differences in molecular folding and glycosylation in the eukaryotic vs. the prokaryotic system [23]. Despite the high sensitivity of our eukaryotic rRBD-based ELISA, it decreased considerably when compared to commercially available kits. However, our ELISA can still provide useful information related to the level of neutralizing protective antibodies in the sera of individuals at a better cost–benefit ratio.

Considering that SARS-CoV-2 RBD binds the human ACE2, allowing virus entrance to the cell, it is conceivable that levels of anti-RBD antibodies could correlate with levels of neutralizing antibodies. Our results showed that anti-RBD IgG antibody levels evaluated by our ELISA test were strongly correlated with SARS-CoV-2 neutralization antibodies evaluated through a microneutralization assay (*p* = 0.0001). This result suggests that our ELISA can be used as a useful surveillance tool to evaluate the levels of protective antibodies in the population, allowing evidence-based decision-making for health authorities. It is worth mentioning that more than six different vaccines have been used in the national campaign against COVID-19 in Mexico [24]. Growing evidence indicates that the titers of protective antibodies following vaccination protocols with different vaccines vary considerably [25,26,27], which emphasizes the necessity of careful follow-up evaluations of the population immunity—hence the importance of counting with quick and low-cost diagnostic procedures suitable for extensively monitoring of the level of protective immunity.

Here, we describe a recombinant RBD-based ELISA, which allows correlation between the levels of total IgG antibodies with the levels of protective, neutralizing antibodies in SARS-CoV-2-positive human sera. As our method has already been validated by agencies in charge of certification, prospects include the extensive use of this tool for the follow-up of immunity in large populations in Mexico.

## Figures and Tables

**Figure 1 diagnostics-12-01629-f001:**
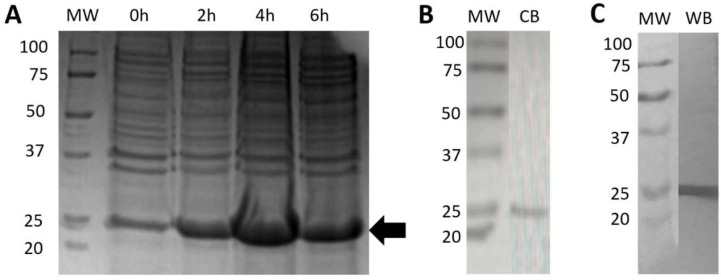
Expression and purification of recombinant bacterial SARS-CoV-2 RBD. (**A**) SDS-PAGE of fractions taken after transformed BL21(DE3) cells were induced to express RBD using IPTG. The arrow indicates the position of the expected recombinant product; (**B**) SDS-PAGE of the purified recombinant RBD stained with coomassie blue; (**C**) Western blot of the purified recombinant RBD using a mouse monoclonal antibody against the poly-histidine tag. Abbreviations: MW, molecular weight markers; CB, coomassie blue; WB, western blot.

**Figure 2 diagnostics-12-01629-f002:**
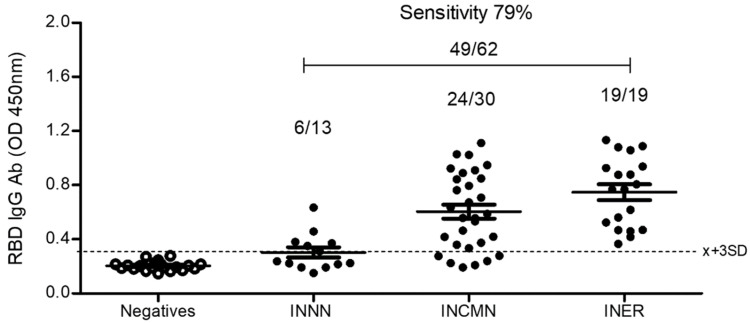
ELISA using bacterial recombinant RBD and sera from SARS-CoV-2 infected and uninfected individuals. Plates were coated with 5 µg/well of purified recombinant RBD. The reaction was evaluated by measuring OD at 450 nm. Sera were diluted 1:200; the dotted line shows 3 standard deviations above the mean of OD for the group of uninfected sera.

**Figure 3 diagnostics-12-01629-f003:**
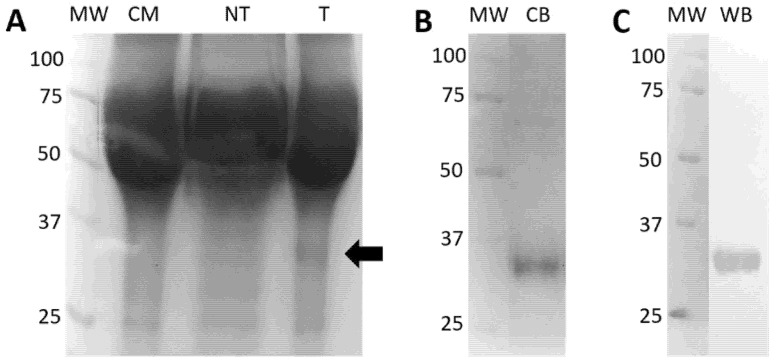
Expression and purification of eukaryotic SARS-CoV-2 RBD. (**A**) SDS-PAGE of fractions taken after HEK293T mammalian cells were transformed using polyjet. The arrow indicates the position of the expected recombinant product; (**B**) SDS-PAGE of the purified recombinant RBD stained with coomassie blue; (**C**) Western blot of the purified recombinant RBD using a mouse monoclonal antibody against the poly-histidine tag. Abbreviations: MW, molecular weight markers; CM, culture medium; NT, non-transformed HEK293T cells; T, transformed HEK293T cells; CB, coomassie blue; WB, western blot.

**Figure 4 diagnostics-12-01629-f004:**
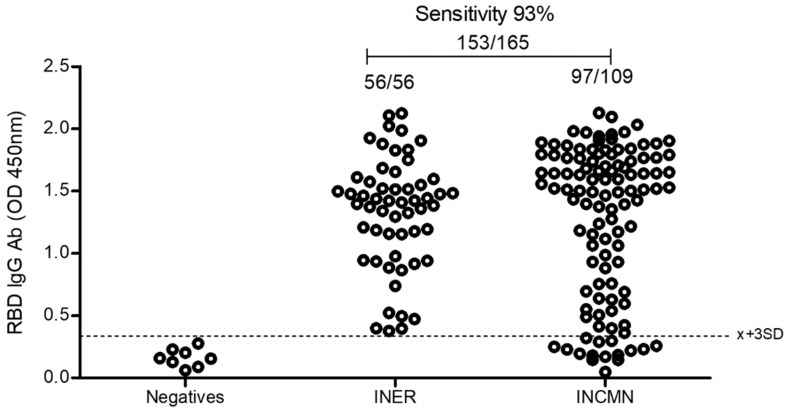
ELISA using eukaryotic recombinant SARS-CoV-2 RBD produced in mammalian HEK293T cells and sera from infected and uninfected individuals. The plates were coated with 50 ng/well of RBD. The reaction was evaluated by measuring OD at 450 nm. Sera were diluted 1:200; the dotted line shows 3 standard deviations above the mean of OD for the group of uninfected sera.

**Figure 5 diagnostics-12-01629-f005:**
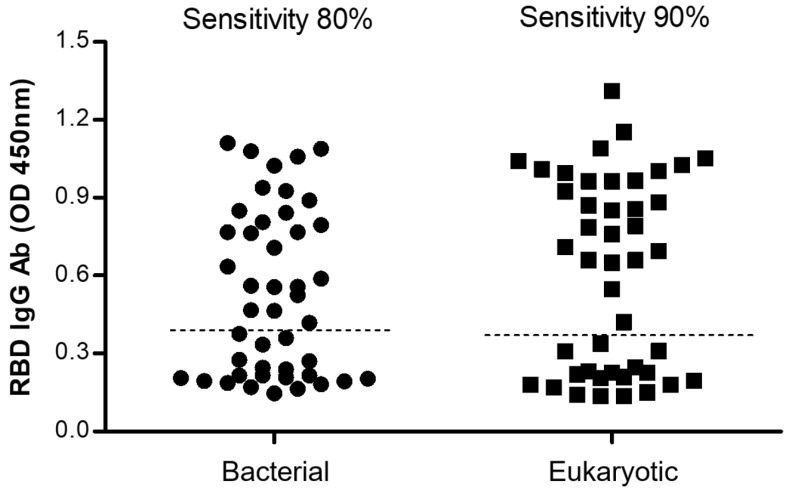
Comparison of ELISA tests using bacterial and eukaryotic recombinant SARS-CoV-2 RBD produced in BL21(DE3) and HEK293T cells, respectively. The dotted lines represent 3 standard deviations above the mean of a panel of negative controls.

**Figure 6 diagnostics-12-01629-f006:**
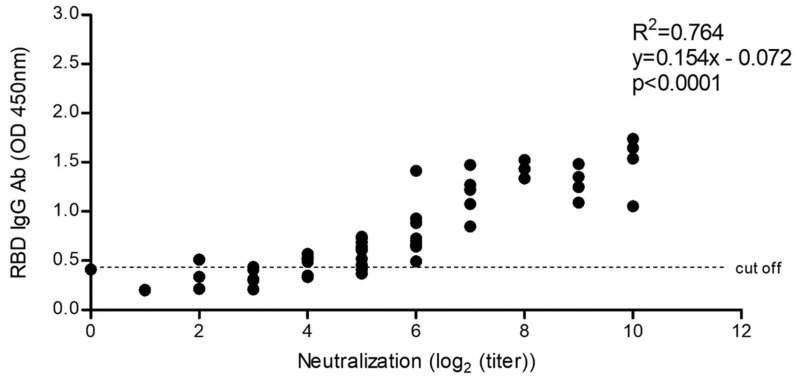
Microneutralization assay of sera samples from vaccinated individuals that were preselected after antibody testing using our eukaryotic rRBD ELISA. The assay was carried out in two-fold serial dilutions of the sera samples added with an equal volume of SARS-CoV-2 virus at an MOI of 0.1 and then added to Vero E6 cell monolayers. The resulting cytopathic effect was visualized every day under the microscope. Positive (with a pre-evaluated lytic capacity) and negative controls (no virus added) were included in the assay.

## Data Availability

Not aplicable.

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
