# Peer review of "An RBD-Based Diagnostic Method Useful for the Surveillance of Protective Immunity against SARS-CoV-2 in the Population"

_diagnostics, 2022, doi:10.3390/diagnostics12071629_

Round 1

Reviewer 1 Report

The paper by Ayon-Nunez described an ELISA-based method for detecting antibody against SARS-CoV-2 in serum samples. RBD was used to coat the ELISA plate. They found that assays using RBD purified from the mammalian culture was far more sensitive than that using RBD from E. coli. They confirmed that their ELISA-based assay can relatively accurately detect anti-SARS-CoV2 antibody in serum, although the sensitive is worse than that of the commercial kits. They claimed that their ELISA-based assay can be performed at cheaper cost than the commercial kits. However, there was no price estimation, making it impossible to compare directly. There are a few additional concerns:

  1. In the ELISA assay, 5 µg of prokaryotic and 50 ng of eukaryotic rRBD were used to coat the ELISA plate. Will the assay sensitivity increase if a higher amount of eukaryotic rRBD is immobilized on the ELISA plate?
  2. What is the expression/purification yields of eukaryotic rRBD? What’s the estimated cost of this ELISA-based diagnostic assay?
  3. The data presented in Figure 6 is somewhat ambiguous as there is no definition of “low, medium and high antibody levels”. Can the data in Figure 6 be linked to that in Figure 5 so the readers can better appreciate the different levels of antibody in serum?
  4. Line 337-338: It is not clear what it means by “decrease in sensitivity …. of 48.4% and 82.3 %”. The measured sensitivity of the commercial kits should be provided.
  5. Line 114: It is unclear what it means by “50% imidazole”.
  6. Line 206: The authors need to specify the stain of SARS-CoV-2 used in the assay.

Reviewer 2 Report

The present manuscript is overall good work, however, it should be clearly indicated in Figures 1B, C, and 3B, C that they are two different gels or the gel figure is composite. Furthermore, in the legend in Figure 2, the abbreviations should be explained (e.g., INNN, INCMN, etc.). Otherwise, the manuscript is well-written and understandable and should be published.

Round 2

Reviewer 1 Report

All my concerns have been adequately addressed.